# Volatilomics of Fruit Wines

**DOI:** 10.3390/molecules29112457

**Published:** 2024-05-23

**Authors:** Tomasz Tarko, Aleksandra Duda

**Affiliations:** Department of Fermentation Technology and Microbiology, Faculty of Food Technology, University of Agriculture in Krakow, ul. Balicka 122, 30-149 Krakow, Poland; aleksandra.duda@urk.edu.pl

**Keywords:** volatilomics, fruit wines, volatile compounds, yeast, analytical methods

## Abstract

Volatilomics is a scientific field concerned with the evaluation of volatile compounds in the food matrix and methods for their identification. This review discusses the main groups of compounds that shape the aroma of wines, their origin, precursors, and selected metabolic pathways. The paper classifies fruit wines into several categories, including ciders and apple wines, cherry wines, plum wines, berry wines, citrus wines, and exotic wines. The following article discusses the characteristics of volatiles that shape the aroma of each group of wine and the concentrations at which they occur. It also discusses how the strain and species of yeast and lactic acid bacteria can influence the aroma of fruit wines. The article also covers techniques for evaluating the volatile compound profile of fruit wines, including modern analytical techniques.

## 1. Introduction

According to the nomenclature, wine is an alcoholic beverage produced by the fermentation of grape juice, while fruit wines are alcoholic beverages made from fruits other than grapes, which are subjected to fermentation but not to distillation. Regardless of the raw material of the fruit, wine is a complex mixture of metabolites derived from the fruit from which it is made, from those produced during fermentation by yeast and bacteria, and, in the case of barrel-aged wines, from the wood from which the barrels are made.

Depending on the region, climate, and growing tradition, the fruits used to make fruit wines vary. In most countries of the world, apple wines are dominant, but a variety of other fruits are also used in the production of fruit wines. In the United States and Canada, berries and stone fruits such as strawberries, plums, peaches, and blackberries are commonly used. In Europe, pears, plums, and berries such as currants, blueberries, and gooseberries dominate. In Asia, citrus and tropical and subtropical fruits are popular, while in Africa, native fruits such as *Parinari curatellifolia* (sand apple, hacha), *Uapaca kirkiana* (mazhanje), and *Ziziphus mauritiana* (masau) are used. These wines typically have an alcohol content ranging from 1.2% to 14%. This category encompasses not only fruit wines but also low-alcohol ‘cider-style’ wines, cryo-extracted fruit wines, fortified fruit wines, and sparkling fruit wines. They can be classified as dry, semi-dry, semi-sweet, or sweet [1,2].

The chemical compounds found in wines can be divided into two categories: volatile and non-volatile components. The first group includes sugars (mainly glucose and fructose), organic acids (mainly malic, tartaric, and citric acids), amino acids (mainly proline and arginine), and secondary metabolites such as anthocyanins, flavonoids, phenolic acids, flavan-3-ols, procyanidins, tannins, stilbenes, and viniferins. The second group, volatile organic compounds (VOCs), affects the quality of wine, in particular determining its aroma. VOCs are mainly formed during fermentation, with the participation of microorganisms, and during the ageing of wine. They include, but are not limited to, alcohols, aldehydes, ketones, lactones, esters, terpenes, norisoprenoids, methoxypyrazines, and thiols (usually less than 300 Da) [3,4].

The consumer’s first contact with a wine is primarily related to the aroma of the beverage, and therefore depends on the volatile compounds present in it. The origin of the volatiles that make up a wine’s aroma is very complex and depends on many factors, including the species of fruit, its chemical composition, the growing region, the climate, the strains of microorganisms used to produce it, and the conditions prevailing during the fermentation and ageing stages [5].

‘Volatilomics’ is a field of science that studies the volatile organic compounds emitted or transformed by various living organisms, using metabolomic tools to characterise the analytes. Historically, volatilomics has been used as a tool for non-invasive medical diagnosis and the early detection of various diseases, by analysing the unique chemical fingerprint created by VOCs secreted by human cells, tissues, and organs. However, volatilomics can cover all living organisms (plants, animals, fungi, bacteria) and can therefore be a useful tool in a wide range of applications [6]. The term ‘volatilome’ has been introduced to describe the volatile compounds found in an organism or food matrix, including those derived from microbial metabolic processes, as well as exogenously derived compounds (organic and inorganic) [7].

This paper discusses the volatilomics of fruit wines and, in particular, the volatile compounds of wines, their origin, and perception by the human sense of smell, as well as the aromas characteristic of fruit wines. The analytical methods used to detect and identify volatile compounds are also of great importance and are presented in this review.

## 2. Volatile Compounds in Wines—Origin and Aromas

The aroma of wines is made up of several hundred different compounds, with concentrations ranging from 0.1 μg/L to 100 mg/L. The final aroma of wine is a combination of volatile compounds derived from the fruit (the so-called varietal aroma or primary aroma), from yeast and bacterial metabolism (fermentative/secondary aroma), and from winemaking processes that occur after fermentation, including ageing (tertiary aroma) [8,9,10]. The varietal aroma includes terpenes, methoxypyrazine, some varietal thiols, etc., and there is considerable variation in aroma components and concentrations between cultivars. The fermentative aroma includes alcohols, esters, aldehydes, and volatile fatty acids components. These components are generated by yeast metabolism during alcohol fermentation and by lactic acid bacteria when malolactic fermentation occurs. During the ageing process, wine components undergo chemical reactions such as esterification or oxidation, to form unique aroma compounds, including oxoles and lactones [11].

### 2.1. Higher Alcohols

Higher alcohols are typically the aromatic molecules that have the most significant impact on the overall aroma of a wine. The final concentration of higher alcohols in wine is primarily dependent on yeast metabolism, as well as other factors such as the wine type and chemical composition. Higher alcohols are produced by yeast, mainly as a result of amino acid metabolism. There are numerous amino acid precursors (Table 1); however, some higher alcohols do not have amino acid precursors. It is therefore assumed that they are formed from intermediates of the tricarboxylic acid cycle (TCA cycle) [12,13].

The higher alcohols most commonly found in wine include isobutanol, isoamyl alcohol, amyl alcohol, isopentanol, 1-hexanol, and phenethyl alcohol. While many of these alcohols have pleasant aromas, such as the honey and rose aroma of phenethyl alcohol or the whiskey flavour of isoamyl alcohol [15], they can also mask the fruity aroma of the wine [16]. When the concentration of higher alcohols in wine is between 111 and 200 mg/L, it enhances the mildness and creates a unique aroma. At 300 mg/L, the higher alcohols produce a pleasant taste. However, if the total concentration of higher alcohols exceeds 400 mg/L, it results in an unpleasant odour, a rough feeling, and can easily cause headaches. This unpleasant odour can dominate the aroma and interfere with the perception of other volatile components of the wine. Therefore, to produce high-quality wine, the most appropriate higher alcohol concentration is to keep it below 300 mg/L [5,11]. Table 2 presents the thresholds of higher alcohols found in wines and the corresponding aromas. It is evident from this table that certain compounds can affect the aroma of wine even at very low concentrations (e.g., methionol, 1-hexanol), while others can only be detected at concentrations several dozen or even several hundred times higher (e.g., benzyl alcohol).

The synthesis of higher alcohols can be controlled by adjusting the fermentation temperature and pH value. A fermentation temperature of 25 °C is most favourable for the production of higher alcohols (reaching concentrations of around 189 mg/L), with the main ones being isoamyl alcohol, 1-propanol, 2-phenylethanol, and isobutanol [17,18]. The lowering of the pH to approximately 4.8 during fermentation has been demonstrated to promote the production of higher alcohols [11,19].

**Table 2 molecules-29-02457-t002:** Higher alcohols in wines, their aromas, and sensory thresholds.

Compound (IUPAC Name)	Chemical Structure	Aroma Descriptor	Threshold [mg/L]	References
Isobutanol(2-methylpropan-1-ol)	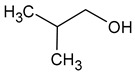	Solvent, chemical alcoholic, malt notes, wineosity notes	40	[14,20]
Isoamyl alcohol(3-methylbutan-1-ol)	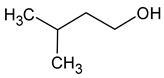	Alcohol notes, nail varnish, solvent amilic notes, malt, whiskey	30	[14,21]
Isohexanol(4-methylpentan-1-ol)	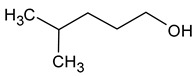	Almond	50	[22]
Tert-amyl alcohol(2-methylbutan-2-ol)	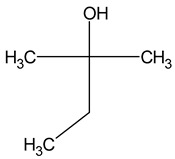	Nail polish, solvent malt	30	[23]
Benzyl alcohol(phenylmethanol)	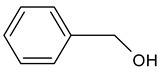	Sweet, floral	200	[24,25]
2-Phenylethanol	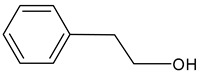	Floral, rose, honey notes, peach notes	10–14	[26]
Hexanol(hexan-1-ol)	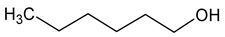	Grass, green	8	[22]
Methionol(3-methylsulfanylpropan-1-ol)	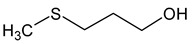	Crushed potatoes	1	[13,14]

Chemical structure based on Pubchem database [27].

### 2.2. Esters

Esters are compounds formed by the condensation of the hydroxyl group of an alcohol with the carboxyl group of an organic acid. They are considered to be one of the most important volatile aroma components in wine, second only to higher alcohols, but they have a lower sensory perception compared to the alcohols [5,28]. Esters are produced naturally by yeast during alcoholic fermentation as a result of yeast metabolism via fatty acids acid acyl- and acetyl-CoA pathways [29].

Ester production is influenced by substrate availability and enzyme activity in yeast. In the case of acetate ester formation, the presence of two substrates, acetyl-CoA and higher alcohol, determines the nature of the acetate ester formed. The availability of acetyl-CoA may play an important role as the main factor limiting this synthesis. Acetyl-CoA levels can be affected by temperature, the addition of fatty acids, nitrogen sources, and the presence of oxygen. The availability of higher alcohols as co-substrates may also be a limiting factor in the synthesis of acetate esters. The overproduction of some higher alcohols also indicates a marked increase in the synthesis of the corresponding acetate ester [13,30,31].

Esters can also be formed as a result of bacterial metabolism. For example, it can be observed that the synthesis of ethyl lactate is closely linked to the concentration of lactic acid produced by malolactic fermentation [32,33,34].

It is also observed that alterations in the concentrations of esters in wines also occur during the course of the ageing and storage of such beverages. New esters are formed and a transesterification reaction, which is dependent on substrate availability, occurs. The loss of fruit and floral aromas during wine storage is associated with the hydrolysis of esters, mainly acetates. It has been established that wines stored at lower temperatures (0–10 °C) undergo less hydrolysis, thus preserving their fruity aromas [35,36].

The total concentration of esters in wine usually exceeds the perception threshold, significantly affecting the final perception of wine by the consumer. Over 150 different esters can be detected in wine, although the majority are present in trace concentrations and do not significantly affect the overall aroma of the wine. Some desirable esters impart pleasant fruity or floral aromas to wines. For example, isoamyl acetate and ethyl hexanoate impart a banana aroma, 2-phenylethyl acetate is associated with a rose aroma, ethyl octanoate and ethyl 2-methylbutanoate with pineapple and strawberry aromas, respectively, while ethyl butanoate and ethyl decanoate have fruity and floral aromas. Other esters are considered highly undesirable if they dominate the aroma of the wine. When the concentration of esters is above 12 mg/L, such as in the case of ethyl acetate, it can negatively affect the wine, imparting a varnish aroma. Furthermore, the primary ester present in wine, ethyl acetate, can also have a suppressive effect on other esters and volatile molecules in the wine, inhibiting the perception of beneficial fruity ethyl esters [5,16]. The aromas and sensory thresholds of wine esters are detailed in Table 3. The most important esters present in wine are the ethyl esters of fatty acids, including ethyl acetate, ethyl butyrate, ethyl hexanoate, ethyl octanoate, ethyl decanoate, hexyl acetate, isoamyl acetate, isobutyl acetate, and phenylethyl acetate [34,37,38].

### 2.3. Volatile Fatty Acids

The majority of the volatile fatty acids present in wine are saturated, straight-chain fatty acids with carbon chain lengths ranging from 2 to 18 carbon atoms. The small groups of branched-chain fatty acids include 3-methylbutanoic acid, 2-methylbutanoic acid, and 2-methylpropanoic acid. The most prevalent fatty acid in wine is acetic acid, with concentrations typically ranging between 150 and 900 mg/L. Acetic acid constitutes over 90% of all volatile acids found in wine [16,42].

The final concentration of volatile acids in wine is influenced by a number of factors, including the species of microorganisms used during fermentation, their strains, and several environmental and physiological factors, such as pH, dissolved oxygen pressure, temperature, and the concentration of yeast nutrients [16,43,44].

Acetic acid is usually produced in small quantities (0.2–0.5 g/L) during alcoholic fermentation, as a by-product of the metabolism of *Saccharomyces cerevisiae*. In amounts greater than 0.8–0.90 g/L, it imparts a vinegary smell and a tart taste to the wine, which is why the wine is considered spoiled. Some fatty acids, such as propionic acid, butyric acid, isobutyric acid, valeric acid, 2-methylbutyric acid, hexanoic acid, octanoic acid, nonanoic acid, and decanoic acid, have an unpleasant aroma that is described as rancid, pungent, greasy, or cheese-like [16,45]. Table 4 presents the fatty acids found in wines and their associated aromas, as well as the sensory detection thresholds for each.

### 2.4. Terpenes

Terpenes are a group of compounds with the general formula (C_5_H_8_)_n_. Wines contain mainly monoterpenes, but also diterpenes, semiterpenes, and sesquiterpenes in smaller concentrations [11]. The most important monoterpenoids found in wines include linalool, (E)-hotrienol, citronellol, geraniol, nerol, (−)-*cis*-rose oxide, and α-terpineol. The monoterpenoids citronellol, geraniol, linalool, nerol, and α-terpineol are the main contributors to the varietal aroma profiles of wines, imparting floral, fruity, and citrus aromas [5,46]. Table 5 lists the main terpene compounds that occur in wines, along with their aromas and perception thresholds.

Grapes contain only 10% of the monoterpenoids in volatile form, while the rest are bound to saccharides by glycosidic bonds. As a result of the vinification process, the aforementioned bonds are hydrolysed by the enzymes present in the fruit, including α-rhamnosidase, α-arabinosidase, or β-xylosidase [47]), although the majority of this hydrolysis is carried out by the glycosidases of yeasts and lactic acid bacteria. It has been shown that yeasts belonging to the *Saccharomyces cerevisiae* species exhibit a significantly reduced potential to release volatile terpenoids, compared to wild yeasts such as *Hanseniaspora uvarum*, *Debaryomyces hansenii*, *Metschnikowia pulcherrima*, *Kloeckera apiculata*, *Pichia anomala*, *Meyerozyma guillermondii*, and *Wickerhamomyces anomalus* [48,49,50]. Among lactic acid bacteria, *Oenococcus oeni*, *Lactiplantibacillus* (formerly *Lactobacillus*) *plantarum*, and *Levilactobacillus* (formerly *Lactobacillus*) *brevis* have also been observed to release free volatile monoterpenoids [9,51,52,53,54,55,56,57,58]. The aforementioned microorganisms exhibit this ability due to the presence of enzymes that hydrolyse glycosidic bonds, including β-d-glucosidase (which releases glucose from the volatile compound), α-l-arabinofuranosidase (catalyses the hydrolysis of interosidic bonds within the arabinofuranosyl glucoside precursor of volatile compounds), α-l-rhamnosidase (enables the sequential hydrolysis of the rhamnosyl glucoside precursors of volatile compounds), and α-apiosidase (cleaves the terminal sugar and apiose, releasing the corresponding β-d-glucoside) [5,47,59,60,61,62].

**Table 5 molecules-29-02457-t005:** Terpene compounds in wines, their aromas, and sensory detection thresholds (ranges based on values reported in various scientific studies).

Compound (IUPAC Name)	Chemical Structure	Aroma Descriptor	Threshold [μg/L]	References
Linalool (3,7-dimethylocta-1,6-dien-3-ol)	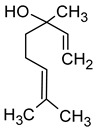	Flowery, fruity	0.006–25	[5,22]
Geraniol ((2*E*)-3,7-dimethylocta-2,6-dien-1-ol)	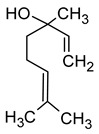	Rose	0.03–32	[5,63]
Citronellol (3,7-dimethyloct-6-en-1-ol)	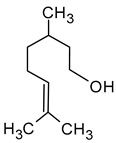	Green lemon	0.008–100	[5,22]
Nerol ((2*Z*)-3,7-dimethylocta-2,6-dien-1-ol)	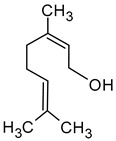	Floral	0.3	[5,64]
β-Damascenone ((*E*)-1-(2,6,6-trimethylcyclohexa-1,3-dien-1-yl)but-2-en-1-one)	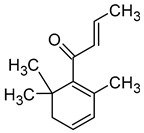	Honey	0.05	[63]
*(E)*-Hotrienol ((5*E*)-3,7-dimethylocta-1,5,7-trien-3-ol)	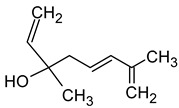	Faint flowery, older flowers	0.11	[5,65]
((−)-*cis*-Rose oxide ((2*S*,4*R*)-4-methyl-2-(2-methylprop-1-enyl)oxane)	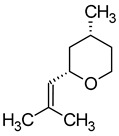	Geranium oil	0.0005	[5,66]

Chemical structure based on Pubchem database [27].

### 2.5. Pyrazines

Pyrazines are a class of nitrogen-containing heterocyclic compounds that are formed by the breakdown of amino acids. The most common pyrazines are 3-isobutyl-2-methoxypyrazine (IBMP), 3-sec-butyl-2-methoxypyrazine (SBMP), and 3-isopropyl-2-methoxypyrazine (IPMP). Although these compounds are present at low concentrations in wines, they may contribute to the perception of an intense aroma of pepper, green pepper, pea, and bean due to their low olfactory detection threshold (Table 6). It has been demonstrated that pyrazine concentrations exceeding 25 ng/L can result in an unpleasant greenish odour in wine. Furthermore, the exposure to strong light has been shown to decrease the concentration of pyrazine [64,67,68].

### 2.6. Thiols

Thiol compounds are typically absent in fruit musts. They are formed with the participation of yeast. Volatile sulphur compounds are usually divided into two categories: highly volatile compounds, most of which are associated with undesirable aroma defects (carbon disulphide, ethanethiol, methanethiol, hydrogen sulphide) and which impart unacceptable aroma notes (e.g., rotten egg, onion, garlic) to wine, and low volatile compounds, which include the main desirable sulphur compounds that contribute to improving the sensory quality of wines [5]. The compounds of the first group are formed during wine fermentation by the assimilative reduction of sulphates by wine yeast (for the biosynthesis of cysteine and methionine). The second group, which is more desirable, consists mainly of 4-methyl-4-sulfanylpentan-2-one (4MSP), 3-sulfanylhexan-1-ol (3SH), and its acetylated derivative 3-sulfanylhexyl acetate (3SHA), which have been described as ‘fruity volatile thiols’ (Table 7). Furthermore, these compounds impart a tropical character to wines, typically resulting in aromas of box tree and blackcurrant buds (4MSP) or passion fruit, grapefruit, citrus peel, gooseberry, and guava (3SH and 3SHA) [5,23,69].

### 2.7. Ageing Aromas

The aroma of wine is the result of an extremely complex interaction between different classes of aromatic compounds and various environmental and biological factors. During the ageing (maturation) process, numerous changes occur in the compounds that shape the aroma of wines. Wine ageing results in the loss of characteristic fruit and floral aromas and the development of new aromas that are characteristic of older wines, or unusual aromas that are associated with wine spoilage. In particular, the concentration of ethyl esters of branched fatty acids changes during the ageing process, and ageing in the lees (mainly yeast cell residues after autolysis) reduces the concentration of volatile compounds that give fruity aromas and increases the concentration of long-chain alcohols and volatile fatty acids. Furthermore, the sediment may also have the capacity to adsorb some of the unpleasant volatile phenols from the wine due to its sorption properties [10,70,71,72].

The ageing of wines in the presence of wood (barrels, wood chips) has a significant impact on the perception of the wine’s aroma. The structure of the wood (porosity, permeability) and its chemical composition (polyphenols, tannins, and volatile compounds) determine the biochemical processes that take place during wine ageing and add complexity to the aroma. Additionally, ageing improves the wine’s stability, enabling the extraction of compounds found in the wood. The wood-derived compounds that give the wine its aroma are mainly furan compounds, lactones, phenolic aldehydes, volatile phenols, and phenylketones. However, if high levels of residual sugars remain during ageing and the molecular form of sulphur dioxide is present at levels below 0.5 mg/L, biological deterioration may occur. Tetrahydropyridines and 4-ethylphenol may produce flavours that are described as “medicinal” or “mousey” [5]. Table 8 presents a list of characteristic compounds that originate from wood and contribute to the aroma of wines.

## 3. Volatile Compounds of Apple Wines and Ciders

The basic raw material for the production of wine is grapes. However, wine can also be made from any fruit that contains the necessary ingredients for its production, namely sugars, acids, tannins, and nitrogen compounds.

Apple fruit is a popular choice for winemakers around the world, with the United States, China, Turkey, and Poland leading the world in apple production. Apple wines and ciders obtained through apple fermentation are particularly popular in Europe, North America, and Australia. The ethanol content of wines and ciders varies, with ciders usually containing less than 8.25% and apple wines containing more than 8.25% [82,83,84,85,86]. The production of apple wines is similar to that of grape wines and involves several stages, including fruit maceration, pressing, must fermentation with malolactic fermentation, ageing and stabilisation, and bottling [83,87,88,89].

The aroma of apple wines and ciders is extremely complex, both qualitatively and quantitatively [90]. The volatile aromatic compounds in apple wine are closely related to the type and concentration of aromatic compounds derived from apples, the fermentation process, and storage conditions. Aromatic compounds are primarily formed during the fermentation process as secondary metabolites of yeast and bacterial metabolism. They comprise esters, higher alcohols, fatty acids, aldehydes, ketones, terpenes, lactones, and others [86,91,92,93]. The quantitative and qualitative composition of volatiles in wines and ciders depends on many factors, including the variety of apple, fruit maturity, and storage conditions [86,94]. The strains of microorganisms employed for fermentation, the conditions of fermentation, including temperature and sugar and alcohol content, as well as the conditions of wine ageing, play a pivotal role in the production of the volatile compounds present in wines and ciders. Various aroma components contribute to the development of the specific and unique flavour pattern of different types of apple wines. In practice, most of the aroma compounds derived from apples are broken down or transformed during their processing, so most of the volatile compounds in apple wine are synthesised during fermentation and maturation. On the other hand, the precursors of aroma compounds formed during fermentation depend on the chemical composition of the fruit and the vinification conditions [86,95]. One of the most important process parameters influencing the profile of volatile compounds is temperature, which affects the rate of nitrogen and sugar assimilation, thus influencing wine quality [96]. Low fermentation temperatures contribute to the formation of significant amounts of floral and fruity aromas but may inhibit yeast growth and slow down fermentation [97]. Temperature determines the development of *Saccharomyces* strains and the effectiveness of their fermentation. The fermentation rate of apple wine was found to be higher at 25 °C than at 13 °C, yet cell viability was higher at 13 °C than at 25 °C [98]. The formation of higher alcohols is also temperature-dependent, with changes in temperature causing alterations in the synthesis of fusel alcohols, from which many esters are formed. In addition to the yeast strains and temperature of apple wine fermentation, the methods and parameters employed during vinification are also important. These include maceration, SO_2_ dosage, nitrogen content, and ageing time [86,99,100,101]. Table 9 shows the profile of volatile compounds in wines and ciders made from different apple varieties. The authors used different strains of *Saccharomyces cerevisiae*.

It is challenging to directly compare the outcome results of various researchers into the composition of volatile compounds. The authors employ disparate methodologies to examine these compounds, which precludes a linear comparison of the concentrations of individual particular aroma components in wines and ciders. The second problem is the way in which the results are expressed and presented. If the results are expressed in terms of concentrations, then their verification is clear and unambiguous. Nevertheless, many researchers present their findings in terms of chromatographic peak areas or in terms of percentages of the peak areas of all the volatile compounds detected, which may not always be identified. Also, the specific volatile compounds evaluated by different researchers may vary. It is possible, nevertheless, to demonstrate the influence of many factors, such as vinification treatments, yeast strain, and the use of lactic acid bacteria, on the composition of volatile compounds in wines and ciders produced and analysed under the same conditions.

An analysis of the data presented in Table 9 reveals a general trend: the concentrations of volatile compounds that contribute to the aroma in ciders are significantly lower than in wines. This is closely related to the different technology of wine and cider production [86,95]. Cider is usually made by fermenting fresh apple juice, without significantly changing the composition of the must. The concentration of ethanol in these products is lower than in wines, which may be sweetened with concentrated apple juice or sucrose to achieve the expected strength. Higher concentrations of ethanol, but also of other alcohols in wines compared to ciders, have a significant effect on the formation of the basic aroma components, namely esters [102]. The lower concentrations of their precursors (alcohols and acids) in cider lead to the synthesis of esters in lower concentrations than in wines.

When analysing the profile of the volatile compounds in the wines (Table 9), it can be observed that the influence of the apple variety on the alcohol concentration is the least significant compared to other groups of compounds—in particular, amyl alcohols and isobutanol. These alcohols are mainly formed from amino acids present in the fruit and depend largely on the method of fertilising the orchards and not on the apple variety [86]. Significant varietal diversity can be seen in the case of esters. The concentration of the most common ethyl acetate ranges from less than 1 mg/L (Ilzer Rose) to almost 55 mg/L (Fuji variety).

The yeast species and strains used for fermentation have a huge impact on the volatile compound profile of wines and ciders [88,91,106,107]. The strains of the species *Saccharomyces cerevisiae* (Johannisberg Riesling and Steinberg) have been demonstrated to influence the profile of volatile compounds in apple wines. The Steinberg yeast strain produced significantly more esters than the Johannisberg Riesling yeast strain when using the same apple variety, but higher concentrations of most alcohols were observed after fermentation with Johannisberg Riesling yeast [91]. Anton et al. [88] showed that the noble strains of the *Saccharomyces cerevisiae* species significantly changed the profile of volatile compounds in ciders, compared to spontaneous fermentation. The authors showed that ciders obtained after inoculation with *Saccharomyces cerevisiae* yeast exhibited lower concentrations of dominant volatile compounds. Significant differences were observed in the concentrations of certain compounds, with values up to 2-fold higher for ethyl acetate, amyl alcohols, 4-ethylguaiacol, and acetate esters; and for 2-phenyl ethanol, the difference was almost 10-fold.

The concentration of ethyl esters in ciders fermented with *Saccharomyces cerevisiae* was often higher than after spontaneous fermentation. In another work [107], the influence of various yeasts, including *Saccharomyces cerevisiae*, *Torulaspora delbruekii*, *Saccharomyces uvarum*, *Hanseniaspora osmophila*, *Hanseniaspora uvarum*, *Starmerella bacillaris*, and *Zygosaccharomyces bailii*, on the profile of cider aroma compounds was studied. The yeast *Saccharomyces cerevisiae* was found to produce the greatest quantity of 2-phenylethyl alcohol, ethyl hexanoate, ethyl octanoate, ethyl decanoate, hexyl acetate, hexanoic acid, and octanoic acid. Low levels of most of the volatile compounds determined were observed after fermentation with *Hanseniaspora uvarum* and *Zygosaccharomyces bailii*. In turn, *Starmerella bacillaris* produced large amounts of 2-butanol, geraniol, and *β*-damascenone. In another study [108], Koji (a rice wine starter culture made from *Aspergillus oryzae*) was used for cider fermentation in addition to the *Saccharomyces cerevisiae* species. The authors made interesting observations. The Koji alone was observed to synthesise a greater number of specific compounds in the fermented must than the *Saccharomyces cerevisiae* strains, including octanoate, ethyl decanoate, ethyl laurate, and hexyl acetate. However, ciders fermented with a mixture of *Saccharomyces cerevisiae* and Koji produced fewer ethyl esters than those fermented with single strains of *Saccharomyces cerevisiae* and Koji. In contrast, the mixture of strains produced more alcohols, including 1-octanol, 1-dodecanol, and amyl alcohols, than single strains.

There is a considerable degree of diversity within a given species of yeast in its ability to synthesise compounds that influence cider aroma, and this depends on the strain used. Cesnik et al. [106] employed 14 strains of *Saccharomyces uvarum* and 6 strains of *Saccharomyces cerevisiae* for cider fermentation. The authors showed that, in certain instances, there were notable discrepancies in the production of volatile compounds between the two species, and, in other cases, between particular strains, even within the same species. On the other hand, the concentrations of ethyl 2-methylbutyrate, E-3-hexenol, and guaiacol produced by all the strains used did not differ significantly. The greatest differences in the amount of volatiles formed were observed for 4-vinylphenol and 4-vinylguaiacol, with concentrations ranging from 18.18 to 561.04 µg/L and from 36.01 to 2121.76 µg/L, respectively. Regarding acetate esters (AEs), no or only minor differences in their production were found between yeast strains belonging to *S. cerevisiae*, although strain 2273 was found to synthesise significantly lower amounts of isobutyl acetate, isoamyl acetate, and hexenyl acetate than other representatives of this species. The differences in AE concentrations within the *S. uvarum* species were found to be greater than those reported for the *S. cerevisiae* species. The lowest levels of AEs were produced by strains 2061 and 2071, while the highest levels among the yeast strains tested were produced by *S. uvarum* strain 2120. The synthesis of ethyl esters of fatty acids (EEFAs) and ethyl esters of branched acids (EEBAs) exhibited less diversity and was largely dependent on the yeast species. The *S. cerevisiae* strains typically produced lower amounts of ethyl isobutyrate and Z-3-hexenol but higher amounts of ethyl 2-methylbutyrate. However, the observed differences were often not statistically significant [106].

The fermentation of wines and ciders with the participation of yeasts and, in addition, lactic acid bacteria also significantly changes the profile of compounds responsible for their aroma [109]. The study showed that malolactic fermentation (MLF) with strain *Leuconostoc mesenteroides* subsp. *mesenteroides* Z25 significantly increased most of the compounds responsible for the aroma of cider. The concentration of isoamyl alcohol increased almost 9-fold, valeric acid 7-fold, acetaldehyde and isosorbide 5-fold, isobutyl alcohol up to 12-fold, and most acids and esters approximately 2-fold. Higher concentrations of 9,12-octadecadienoic acid, dihydroxyacetone, and acetamide were observed in ciders that had not undergone MLF but only alcoholic fermentation. Similar results were obtained in the study by Chen et al. [105], which used *L. plantarum* strains in addition to alcoholic fermentation. It was shown that the concentration of esters in ciders after lactic fermentation increased 1.2–1.5-fold, depending on the bacterial strain used and the duration of fermentation. The highest increases in concentration were observed for 4-hydroxybutyrate (4-fold), methyl *trans*-cinnamate (7-fold), and phenylphosphoric acid (3-fold). MLF also contributed to an increase in the concentration of higher alcohols in ciders. Particularly large increases were observed for phenethyl alcohol (14-fold), diglycerol, and estriol (4-fold) [105].

Ruppert et al. [83] investigated the impact of the maceration of apple pomace during wine vinification on the profile of volatile compounds. They observed an increase in the concentration of butanoic acids 1-pentanol, 1-hexanol, 1-heptanol, 1-octanol, and benzyl alcohol after maceration, whereas the amount of the hexanoic acid 3-ethoxy-1-propanol decreased. Maceration usually promoted the formation of ethyl esters in apple wines, although the concentration of acetate esters was observed to decrease. Furthermore, maceration led to a significant increase (2–4-fold) in the concentration of terpenoids in apple wines.

## 4. Volatile Compounds of Other Fruit Wines

In addition to apple wines, a wide range of fruits can be a raw material for the production of fruit wines. The resulting fruit wines typically exhibit a distinctive aroma, characteristic of the fruit used, and contain precursors of aroma compounds that are formed during the fermentation and ageing of the wines.

One of the most popular fruit wines is cherry wine. It is consumed in many regions of the world, mainly in Asia (especially China) and Europe [110,111]. The aroma of cherry wines is composed of hundreds of volatile compounds, including those derived from the cherries themselves and those resulting from the wine production process, including fermentation by yeast and the ageing process [112]. Table 10 shows the characteristic aromas of cherry wines fermented with the yeast *Saccharomyces cerevisiae*.

The aroma profile of fruit wines is influenced by both the quality of the raw material and fruit varieties, but primarily by vinification conditions. The wines listed in Table 10 have been produced from various species of cherry fruit, including May Duck, Early Richmond, and Lapins, and also include wines that were purchased from various shops and originate from different countries, with no clear indication of the cherry variety. All of them were subjected to standardisation of acidity and sugar content and to fermentation with the yeast *Saccharomyces cerevisiae*. A number of factors influenced the results obtained, but certain trends in the synthesis of aromatic compounds characteristic of cherry wines can be observed. First of all, relatively low concentrations of organic acids were found, with acetic acid being the most prevalent (Table 10). From an oenological point of view, cherry must is highly acidic, with 7–10 g/L of malic acid. It may therefore be necessary to reduce this acidity according to requirements. Cherries often contain insufficient quantities of sugars for wine fermentation and require chaptalisation, which can lead to a reduction in acidity [112,113]. High acidity is a limiting factor in yeast-mediated acid synthesis. Compared to apple wines (Table 9), the production of acetic acid by yeast was found to be approximately 10 times lower. The lower concentration of acetic acid in cherry wines results in the formation of small amounts of acetate esters, which are present in much lower concentrations than in apple wines. The dominant esters in cherry wines are ethyl esters of acids present in must and produced by yeast [114]. It is noteworthy that ethyl lactate was found in all cherry wines tested [113,115]. This compound has rarely been determined in apple wines [86]. Also noteworthy are the relatively low concentrations of fusel alcohols and methanol in cherry wines [116].

**Table 10 molecules-29-02457-t010:** Compounds that contribute to the aroma of nine various cherry wines [mg/L].

Compounds	W1	W2	W3	W4	W5	W6	W7	W8	W9
Ethyl acetate	1.60	1.40	5.86	0.13	8.22	12.88	6.25	33.07	1.68
Ethyl 3-methylbutanoate	0.01	0.04	nd	nd	na	0.03	nd	na	na
Isopentyl acetate	0.60	na	na	na	na	na	na	na	0.04
Ethyl pentanoate	0.02	na	na	nd	na	na	na	na	na
Ethyl hexanoate	0.09	0.09	0.14	0.46	0.52	0.17	0.29	0.24	0.38
Hexyl acetate	0.17	0.11	na	na	0.22	0.02	na	0.01	nd
Ethyl 3-hexenoate	0.08	0.47	na	na	nd	na	na	na	na
Ethyl lactate	5.08	5.08	2.72	7.97	0.60	1.25	7.44	0.34	0.17
Ethyl octanoate	0.76	0.75	0.66	na	0.46	0.37	0.51	0.82	na
Ethyl decanoate	0.18	0.06	na	na	0.49	0.19	na	0.10	0.79
Ethyl benzoate	0.47	0.13	na	nd	0.11	0.39	na	0.32	0.01
Diethyl succinate	0.08	0.08	nd	nd	0.79	0.78	na	0.07	0.45
Ethyl 2-phenylacetate	0.01	0.01	na	na	0.10	0.02	na	na	0.02
2-Phenylethyl acetate	0.03	0.22	8.13	na	na	na	0.78	0.01	nd
1-Propanol	29.01	22.96	2.74	na	0.86	22.18	1.41	8.53	0.72
2-Methylpropanol	17.96	12.95	12.66	nd	nd	1.89	0.62	na	na
1-Butanol	0.75	0.55	0.74	0.33	0.31	na	0.68	na	0.52
3-Methylbutanol	49.46	12.71	9.98	102.32	1.86	13.87	10.84	14.64	na
3-Methyl-3-buten-1-ol	1.53	na	na	na	na	na	na	na	na
1-Pentanol	0.01	0.06	na	na	na	na	na	na	na
1-Hexanol	1.76	0.66	na	5.89	0.309	0.46	0.66	0.50	0.80
2,3-Butanediol	0.26	0.05	na	na	na	na	na	na	0.05
1-Octanol	0.03	0.03	na	na	na	na	na	na	0.11
Benzyl alcohol	17.08	5.09	3.45	3.65	1.09	18.51	2.05	17.82	2.12
2-Phenylethyl alcohol	16.35	6.76	4.96	1.48	0.58	8.38	2.42	9.18	0.69
Acetic acid	35.18	25.15	na	9.81	5.28	6.62	5.35	3.73	2.10
2-Methylpropanoic acid	0.85	1.42	2.28	na	na	na	na	na	na
Butanoic acid	0.28	0.11	nd	0.06	nd	na	0.15	na	0.20
3-Methylbutanoic acid	0.40	0.42	nd	na	na	na	na	na	na
Hexanoic acid	18.17	4.92	3.69	0.29	nd	0.61	1.83	1.17	1.15
Octanoic acid	5.36	3.59	3.25	0.39	na	0.98	0.39	0.11	0.51
Decanoic acid	3.97	2.22	na	na	na	na	na	na	2.22
Benzaldehyde	0.12	0.58	2.25	7.37	93.78	7.50	25.41	2.31	3.75
Linalool	0.02	0.15	0.03	na	na	na	na	0.15	0.37
α-Terpineol	0.02	0.02	0.09	na	na	0.26	na	0.22	0.02
β-Citronellol	0.37	na	na	16.80	na	0.05	na	0.05	0.10
β-Damascenone	0.05	1.20	0.07	na	na	na	na	na	na

Legend: W1 [112]; W2 [114]; W3 [117]; W4 [111]; W5 [118]; W6 [115]; W7 [119]; W8 [120]; W9 [116]. na—not analysed, nd—not detected.

Xing et al. [121] show that the utilisation of *Torulaspora delbrueckii* yeast alters the profile of volatile compounds in cherry wines. In comparison to fermentation with *S. cerevisiae*, an increase in the synthesis of most esters was observed, especially acetates and some aldehydes, with the exception of benzaldehyde, of which *S. cerevisiae* synthesised almost twice as much as *T. delbrueckii*. The formation of volatile alcohols was not significantly different. However, other researchers [114] found no significant effect of the *T. delbrueckii* species on the synthesis of esters and only slightly lower amounts of alcohols. Sun et al. [122] revealed no significant effect of *Torulaspora delbrueckii* on the formation of aroma compounds (except for a smaller amount of acids) compared to *Saccharomyces cerevisiae*. However, sequential inoculation of the must with a mixture of cultures of these microorganisms resulted in an increase in the synthesis of esters and alcohols, with a notable enhancement in the production of those responsible for the floral aroma of cherry wines (2-phenylethanol and its esters). Researchers have conducted several experiments to assess the impact of lactic fermentation on the sensory profile of cherry wines. Sun et al. [117] have demonstrated that autochthonous *Lactobacillus plantarum* has a positive effect on the sensory evaluation of cherry wines. The authors showed an increase in the concentration of esters, especially ethyl acetate and ethyl lactate, fusel alcohols, and terpenoids. Nevertheless, other research [110] has shown that the strain of lactic acid bacteria employed is of significance with regard to the positive sensory characteristics of cherry wines. Some cultures of *O. oeni* were observed to possess the capacity to produce fruity aromas and linalool, thereby imparting fruity and floral notes to wines. The cherry wines fermented with endemic lactic acid bacteria exhibited higher levels of hexanoic acid and benzaldehyde, resulting in green plant parts and almond odours, which worsened the results of the sensory analysis. An interesting comparison of the quality of the aroma profile of cherry wines from disparate price ranges was made by Xiao et al. [120]. The authors demonstrated that the most expensive wines exhibited the highest concentration of most esters and alcohols, while the mid-range wines exhibited the lowest. However, the sensory evaluation results were the least favourable for the cheapest wines. According to the authors, this is due to the presence of undesirable odour compounds in average and expensive wines at sub-threshold concentrations, while in cheap wines, they were present at above-threshold (noticeable) concentrations. In turn, cheap wines contained low concentrations of compounds that contribute to a fruity and floral aroma.

In addition to the most popular fruits, such as apples and cherries, fruit from the plum family is also used in the production of fruit wines. A diverse array of plum species and varieties are cultivated globally. These include Hungarian plums, apricots, peaches, mirabelle plums, greengages, and lesser-known species such as the Japanese plum (*Prunus mume*) [123,124,125,126,127]. The production of plum wines necessitates additional procedures, including the pitting of fruit, the regulation of the content of fermentable sugars and acidity, and a more challenging clarification process [127]. Table 11 presents the concentrations of aroma compounds in plum wines of different varieties.

It can be observed that the profile of volatile compounds is markedly influenced by the specific plum variety. Plum wines are distinguished by significantly higher concentrations of most esters and alcohols compared to apple wines. Apricot wines are particularly rich in aroma-active compounds, containing approximately twice as many as peach wines and up to ten times more than plum wines. The profile of Japanese plum fruit wines has been studied by Liu [126], but the results were presented in terms of the percentage of chromatographically identified compounds, rather than their concentrations. Consequently, the results cannot be compared with those of other researchers. The authors identified 55 volatile compounds in Japanese plum wines, including 28 esters, 11 alcohols, 4 alkenes, 3 aldehydes, 3 acids, 2 alkanes, 2 phenols, 1 ketone, and 1 naphthalene. The main volatile component was 2-hydroxypropanoic acid ethyl ester (70.511%). Other notable compounds include ethyl acetate, isoamyl alcohol, 3-methylbutyl acetate, and ethyl decanoate. With the exception of the ethyl ester of hydroxypropanoic acid, these compounds were also present in high concentrations in other plum wines (Table 11).

The berry group, which is composed of various fruits, including grapes, blueberries, raspberries, mulberries, and strawberries, has recently gained significant interest among producers and scientists due to the anthocyanin content of these fruits, which plays a crucial role in determining the colour and unique aroma of the wines produced from them. The production processes for wines made from these fruits do not differ significantly from the production of grape wines. However, the must may require additional processing typical of fruit other than grapes. This includes regulating the acidity and the level of fermentable sugars, the concentration of which may be too low to produce high-quality wines [128,129,130].

Table 12 presents a profile of the compounds that contribute to the aroma of selected berry wines.

In this group of wines, the majority of volatile compounds and the highest concentrations were found in strawberry wines. Nevertheless, despite the high concentrations of these compounds, especially fusel alcohols, strawberry wines were not rated very highly by consumers. The aroma of wines is largely determined by esters, and in this case, the fact of their lower concentration than alcohols had a much greater impact on the sensory experience [11]. Raspberry wines contain higher amounts of compounds that are particularly valuable from the consumer’s point of view, such as 2-phenylethyl alcohol, hexyl acetate, and slightly less 3-methylbutanol. However, in higher concentrations, these compounds give the wines an unpleasant aroma [28]. It should be remembered that the quality of wines is determined not only by their aroma but also by their colour. In the case of red wines, this depends on the quantity of anthocyanins present in the fruit skins. For the berries described, the highest levels of anthocyanins are found in blueberry and mulberry [123,128].

Li et al. [131] demonstrated that lactic acid bacteria with active β-glucosidase can influence the profile of volatile compounds in wines by releasing glycoside-bound flavours. This enzymatic activity in *Lactobacillus brevis* significantly increased the concentrations of terpenoids and 2-phenylethyl acid esters, which are responsible for the desired floral aromas. Similar relationships have also been demonstrated for grape wines [132]. Huang et al. [123] found that the use of yeasts other than *S. cerevisiae*, such as *H. uvarum*, for fermentation enhances the production of esters and other compounds responsible for the aroma of blueberry wines. This simultaneously reduces the proportion of alcohols, especially fusel alcohols.

Citrus fruits and pineapples are cultivated in Asia, Africa, and America, and are also used as raw materials for wine [133,134]. They belong to different families, *Rutaceae* and *Bromeliaceae*, respectively. However, in Asian countries, particularly in China, wines are made from citrus fruits, with oranges and pineapples being the most common. Wines made from these fruits exhibit a distinctive aroma, characteristic of the species employed. The taste and aroma of orange juice is one of the most widely recognised in the world. However, the fermentation process associated with the activity of yeasts and bacteria leads to changes in the sensory profile of the resulting wines [135,136]. The use of pineapple in winemaking is linked to two main aspects. Firstly, pineapple is a seasonal fruit, and it can be treated as a raw material for wine in order to avoid losses. The second reason is the low availability of typical wine fruits, especially in North Africa, which had led to experiments in the production of pineapple wines [134,137]. Table 13 shows the profile of the compounds responsible for the aroma of citrus and pineapple wines.

The profile of the compounds responsible for the aroma of orange wines has been a subject of investigation by various researchers, and the results have demonstrated considerable diversity. The volatile compounds of interest have been studied employing a variety of methodologies, making direct comparison challenging. However, it can be noted that the dominant compounds are amyl alcohols and isobutanol, which are formed during must fermentation. Significant concentrations of 2-phenylethyl alcohol, which is desirable in wines for its fruity aroma, are also observed. The group of esters is dominated by ethyl esters. However, it has been demonstrated [136] that the primary influence on the aroma of orange wines is exerted by esters (ethyl hexanoate, isoamyl acetate, ethyl caprylate, phenethyl acetate, and ethyl caprate) and, to a much lesser extent, by higher alcohols (1-pentanol and 2-phenethyl alcohol), citronellol, and octanoic acid. As a result of olfactometric tests, Selli et al. [138] identified ethyl butanoate, 3-methyl-1-pentanol, linalool, γ-butyrolactone, and 2-phenylethanol as the compounds mainly responsible for the aroma of orange wines. The authors [136] emphasised the crucial role of the yeast used for fermentation. The aforementioned compounds were found to be characteristic of the yeast *Saccharomyces cerevisiae*.

Non-*Saccharomyces* yeasts are capable of synthesising other components. In the case of the different *Hanseniaspora* strains, ethyl caprylate, ethyl hexanoate, and isoamyl acetate were the main contributors to the aroma of orange wines, with traces of 2-phenylethanol and octanoic acid. In contrast, the yeast *Barnettozyma californica* was found to predominantly contribute to the aroma of the wines through the synthesis of terpenoids, particularly linalool and citronellol. Esters, in particular ethyl esters, were found to be less significant in influencing the bouquet of the wines. For the production of citrus (Ponkan) wines, it is recommended to use mixed cultures, comprising *S. cerevisiae* and *H. uvarum* or *T. delbrueckii*. The resulting wines exhibit enhanced aroma characteristics [136]. In the production of pineapple wines, isoamyl acetate and ethyl octanoate are crucial in shaping the aroma of the wines, regardless of the yeast species employed, whereas the alcohols present in the wines do not play any role [137].

In selected regions, wines are also made from fruits that are rare, exotic, and used by a limited number of people. These fruits can also be used to produce wines with distinctive aromatic characteristics. Table 14 provides a summary of the volatile compound profile of select unique wines. The composition of the compounds responsible for the aroma of wines varies greatly depending on the type of fruit from which they are made. However, in all cases, the authors emphasise that the beverages they produce have gained consumer acceptance. The fruits described in Table 14 differ greatly in their structure, in the concentration of the compounds that give the wine its aroma, and in their precursors used by the yeast. Many of these wines (W25–W29) are characterised by an extremely low alcohol content (4–6.5%), due to the low concentration of sugars present in the fruit [139]. During the production of these wines, chaptalisation was not used to increase the ethanol content of the final product. The low ethanol concentration may result in significantly lower ethyl esters levels compared to the W23 wine (made from *Rosa roxburghii*), which contained approximately 10% ethanol [123]. Nevertheless, despite the low ethanol concentration, wines produced from these exotic fruits contained volatile compounds (alcohols, monoterpene compounds, and ethyl esters) that contributed to the formation of aromas described as fruity, green apple, banana, sweet, citrus, vanilla, rose, and honey. Passion fruit wines exhibited a high concentration of the analysed volatile compounds that form aromas, as shown in (Table 14). The wines in question exhibited a total acidity that was more than three times higher than the standards permitted in Brazil. Additionally, they contained a higher concentration of most of the acids analysed than that observed in other exotic fruit wines [140] and also a higher ethyl acetate content (40.83 mg/L) (Table 14). Melon wines were those in which the acid content prior to fermentation was insufficient and was required to be corrected by acidification [141]. These wines contained low concentrations of most aroma-forming compounds, especially fusel alcohols. However, due to the presence of precursors, components (2-methyl-1-butyl acetate, ethyl palmitate) not observed in other exotic fruit wines were present. The authors [141] posit that the production of wine from melon can be a very good direction for the use of fruit not meeting quality parameters for direct consumption.

The profile of aroma-forming compounds in exotic fruit wines, as in other wines, is highly dependent on the species and strain of yeast used [123,142]. The profiling of volatile compounds by selected yeast strains, especially in mixed cultures, has been demonstrated in the fermentation of pomegranate wine [143]. The authors demonstrated that the yeast strain used can be adapted to a specific type of fruit used for winemaking, producing organoleptically desirable beverages. Furthermore, they demonstrated the significant influence of the fermentation method and the use of immobilised yeast, with wines produced using this method fermenting faster but containing a much lower concentration (on average by 30%) of volatile compounds responsible for aroma [144].

**Table 14 molecules-29-02457-t014:** Aroma composition of exotic wines [mg/L].

Compounds	W23	W24	W25	W26	W27	W28	W29	W30	W31
Ethyl acetate	22.50	0.05	na	na	na	na	na	40.83	0.67
Isopentyl acetate	101.73	na	na	na	na	na	na	9.24	na
Ethyl butyrate	na	na	0.02	0.13	0.12	0.01	0.01	na	0.04
Ethyl hexanoate	103.96	na	na	na	na	na	na	1.62	0.16
Hexyl acetate	28.81	na	na	na	na	na	na	0.11	na
Ethyl 3-hexenoate	na	na	<0.01	0.01	0.01	0.01	<0.01	0.04	na
Ethyl lactate	na	na	0.20	0.26	0.01	0.41	0.01	na	na
2-Methyl-1-butyl acetate	na	na	na	na	na	na	na	na	0.19
Ethyl octanoate	455.74	0.13	<0.01	0.01	0.13	<0.01	0.00	na	1.27
Ethyl decanoate	519.80	na	na	na	na	na	na	0.26	0.60
Ethyl caprylate	na	na	na	na	na	na	na	4.67	na
Diethyl succinate	na	1.94	1.74	0.55	0.37	2.19	0.17	0.16	na
Ethyl palmitate	na	na	na	na	na	na	na	na	0.38
Ethyl 2-phenylacetate	2.00	na	na	na	na	na	na	na	na
2-Phenylethyl acetate	na	0.04	0.06	0.06	0.01	0.04	0.03	0.47	0.07
1-Propanol	na	na	na	na	na	na	na	na	na
2-Methylpropanol	na	na	na	na	na	na	na	88.92	0.11
1-Butanol	na	0.04	<0.01	0.10	<0.01	0.02	<0.01	na	na
3-Methylbutanol	150.00	24.17	na	na	na	na	na	389.94	1.43
3-Methyl-3-buten-1-ol	na	na	0.02	0.13	nd	nd	0.01	na	na
1-Pentanol	na	0.08	0.17	nd	nd	<0.01	nd	na	na
1-Hexanol	8.00	na	<0.01	0.03	0.04	0.01	<0.01	na	0.07
1-Octanol	na	na	<0.01	0.01	<0.01	0.01	<0.01	0.12	0.05
Benzyl alcohol	na	na	0.01	0.01	0.01	0.02	0.01	0.77	na
2-Phenylethyl alcohol	50.00	17.67	nd	nd	0.02	nd	nd	2.52	0.33
Acetic acid	3.00	na	na	na	na	na	na	na	0.70
3-Methylbutanoic acid	na	na	0.14	0.33	0.11	0.02	0.03	na	na
Hexanoic acid	3.00	0.38	0.54	0.63	0.24	0.15	0.39	13.97	na
Octanoic acid	20.50	0.81	0.01	0.43	0.43	0.45	0.51	54.41	0.12
Decanoic acid	15.00	0.14	nd	0.03	0.03	0.01	0.01	118.78	na
Benzaldehyde	na	na	na	na	na	na	na	1.26	na
Linalool	na	na	nd	<0.01	0.19	0.02	0.01	4.66	na
α-Terpineol	na	na	0.01	0.21	0.01	0.04	0.28	1.86	na
β-Citronellol	na	na	nd	<0.01	0.01	nd	nd	0.53	0.04

Legend: W23—Rosa roxburghii [123]; W24—Cagaita [144]; W25—Cocoa, W26—Cupuassu, W27—Gabiroba, W28—Jaboticaba, W29—Umbu; W25–W29 [139]; W30—Passion Fruit [140]; W31—Melon [141]. na—not analysed, nd—not detected.

All of the fruit wines were dominated by amyl alcohols and isobutanol. However, the concentration of these compounds was found to be highly dependent on the species of fruit used for fermentation. In wines produced from plums and berries, the content of amyl alcohol and isobutanol was found to be significantly higher than in wines from other fruits, with some instances demonstrating a 10-fold increase. Other higher alcohols, such as 1-octanol, also exhibited a dominant presence in wines derived from berries. A notable degree of variability was observed in the alcohol profiles of blueberry, apple, and cherry wines. However, cherry wines exhibited approximately twice as many alcohols as apple wines, which may be attributed to the higher sugar concentration in the fruit. Compared to wines made from other fruit species, sour cherry wines were characterised by a high diversity and a high concentration of volatile acids. The presence of various acids and alcohols resulted in a considerable variation in the profile of esters, particularly ethyl esters, in these wines. In cherry wines, the occurrence of ethyl lactate in relatively high concentrations is a defining characteristic. In contrast, high concentrations of ethyl acetate are typical of plum wines (300 mg/L on average), reaching values that are approximately 25 times higher than in apple wines and 45 times higher than in cherry wines. Additionally, cherry wines exhibit elevated levels of terpenoids. Wines derived from citrus, pineapple, and exotic fruits exhibit a markedly less diverse volatile compound profile, with concentrations of these components often being considerably lower than those observed in cherry, plum, or apple wines. Nevertheless, the concentrations of certain acids, such as decanoic acid in wines derived from exotic fruits (e.g., Rosa roxburghii and passion fruit), were found to be particularly elevated.

A comparison of the composition of volatile compounds in grape wines with those in fruit wines is almost impossible. The profile of the compounds that contribute to the aroma of grape wines is influenced by a multitude of factors, including the variety of grapes used, the strain of yeast employed, the region of origin, and climatic conditions. As a result, it is challenging to derive an average value for these compounds. However, certain universal trends can be observed. All wines are dominated by amyl alcohols, isobutanol and, in the ester group, ethyl acetate and ethyl lactate [4,25,145,146,147,148,149]. Red wines have higher concentrations of most volatile compounds, especially alcohols and lactones. However, in some cases, white wines have higher amounts of quats and some esters [25]. A comparison of results from other works [146,148] reveals that all the white wines tested (12 samples) had significantly lower concentrations of all volatile compounds compared to Cabernet Sauvignon, Merlot, and Cabernet Gernischt wines. However, discrepancies in the composition of selected volatile compounds in wines from the same grape variety but from a different region can yield differences of up to 20-fold [146,150]. Furthermore, the yeast strain employed for fermentation of the same must can result in up to 50-fold differences in volatile compound content [145]. For the purpose of comparison, specific wines should be selected and juxtaposed, but generalised conclusions cannot be drawn for other wines.

## 5. Methods of Analysis—Volatilomics

The term ‘volatilomics’, in addition to the study of the composition of volatile compounds (in the context of this review, affecting the aroma of wines), also refers to methods for their laboratory analysis. This analysis includes sample preparation, extraction methods, analytical conditions, instrumentation, and multivariate data analysis. Gas chromatography is the most commonly used technique for the identification and quantitative analysis of volatile compounds in wines, including fruit wines. In some cases, the complexity of fruit wine systems necessitates the development of chromatographic methods tailored to these matrices, which may require the use of advanced instrumentation and/or novel analytical techniques [7,151,152].

### 5.1. Sample Preparation

Prior to the actual analysis, the wine samples are prepared in an appropriate manner in order to purify the samples and concentrate the analytes. In the case of wines, a two-stage analysis is employed, whereby volatile compounds present in higher quantities are analysed without concentration and those present in lower quantities following concentration. The extraction of volatile compounds is primarily conducted through liquid–liquid extraction and solid-phase microextraction (SPME). Liquid–liquid extractions can be performed, for example, with dichloromethane, which is then concentrated with nitrogen [152]. The ratio of sample to extractant varies between the various papers. Selli et al. [135] mixed 100 mL of a wine sample with 40 mL of dichloromethane for 30 min at 4 °C, followed by centrifugation (9000× *g*). Subsequently, the extract was collected and concentrated to a volume of 0.5 mL, which was applied to a gas chromatograph column. In a separate study, 0.4 mL of dichloromethane was added to 8 mL of wine and stirred for 15 min. The extract collected after centrifugation (5118× *g*) was introduced to the gas chromatograph without concentration. In contrast, the work of Sousa et al. [153] employed a distinct extractant, trichloroethylene. The authors initially diluted the wine sample with water (1:2), then alkalinised it with 23% K_2_CO_3_ to a pH greater than 10 and centrifuged (1690× *g*) after extraction.

The most prevalent method employed in modern analytical practice is solid-phase microextraction (SPME), which facilitates the concentration and purification of samples prior to their introduction into a gas chromatograph. The technique is rapid, simple, inexpensive, and solvent-free. The procedure involves the placement of a sorbent-coated fibre in a supersurface phase or directly in a liquid and heating it for a suitable period of time. Volatile compounds are bound to the sorbent. For desorption, the fibre is placed in the injector of the gas chromatograph and subjected to high-temperature evaporation (e.g., 250 °C), before being introduced to the chromatography column. Various SPME sorbents are used, including PDMS—polydimethylsiloxane, DVB—divinylbenzene, and CAR—carboxene [151,154,155,156,157].

In a study by Kim and Park [158], PDMS/DVB fibre was employed to extract volatile compounds from blackcurrant wines. The researchers added 5 mL of wine to a 10 mL vial, introduced the fibre, and heated it for 30 min at 60 °C. For the extraction of volatile compounds from strawberry wines, PDMS fibres were employed, with extraction either at 30 °C for 30 min [159] or 50 °C for 30 min [160]. In studies that used DVB/Carboxen/PDMS fibre after placing it in a vial of wine, the sample was heated for 60 min at 40 °C [157] or 15 min at 80 °C [151]. In turn, the SPME extraction of wine at 45 °C, for 45 min was used in the work of Welke et al. [161].

Equipping the laboratory with an automatic SPME fibre extraction system coupled to a chromatograph (e.g., Gerstel) enables the significant acceleration of analyses and eliminates human error during determinations [162]. In order to enhance extraction efficiency, the addition of NaCl to the samples is a common practice [151,160,163,164]. In addition, internal standards are introduced into the vials prior to extraction. Depending on the wine under investigation, the authors used a variety of standards, such as 3-acetate [161], 2-acetate [156], 4-methyl-2-pentanol [157], acetone d6, 4-methyl-3-penten-2-one d10, acetanal d16, 4-fluorobenzaldehyde [163], and cyclohexanone [151].

Another technique is the introduction of the gas phase of the wine onto the gas chromatograph column. This is achieved by heating a vial (20 mL) containing a sample (1 mL) to 100 °C for 10 min, after which the volatile compounds are fed to the chromatograph column [164].

### 5.2. Analytical Methods

The chromatographic analysis of wines involves the separation of volatile compounds on appropriate columns and their subsequent detection and quantification by gas chromatography (GC) [7]. Flame ionization detection (FID) is the most commonly used detector for the analysis of volatile compounds. However, the sensitivity of this method is insufficient to identify and quantify many aroma components that occur in very low concentrations [152]. Less commonly used detectors, such as flame detectors, can be employed to evaluate specific wine components. For instance, photometry (FPD) and pulsed flame photometry (PFPD) can be used to identify sulphur compounds, while nitrogen–phosphorous detection (NPD) can be used to determine pyrazides from wood [165].

The quadrupole mass spectrometer (MS) is the second most popular detector for determining volatile compounds in wines, including fruit wines. Its sensitivity is much higher than that of the FID detector, and it is widely used [151,157]. However, wines are a complex mixture of volatile compounds with varying chemical classes, polarities, and molecular weights. This can cause coelutions when using GC-MS. The presence of two or more compounds that coelute can hinder the accurate identification and quantification of volatile compounds. To eliminate this disadvantage, comprehensive two-dimensional gas chromatography with mass detection (GC×GC/MS) can be employed [166,167]. This technique utilises two GC columns coated with distinct stationary phases that are connected in series. The enhanced separation capability of GC×GC compared to conventional GC-MS results in an improved accuracy of MS analysis, which is advantageous for compound identification and quantification [168].

The triple-quadrupole mass spectrometer (QqQ-MS) is a widely recognised detector in the scientific community. The instrument enables the separation, identification, and quantification of volatile components in wines, even at concentrations below 1 ppm [140,163]. The utilisation of high-resolution mass spectrometry with quadrupole time-of-flight measurement (Q-TOF-MS) enables the attainment of an enhanced resolution and sensitivity in chromatographic measurements [154,156]. The development of this technology is the Orbitrap-MS, which allows one to achieve a sensitivity at the femtogram level and which can be an excellent method for determining volatile aromatic compounds in fruit wines [165].

The primary limitation of chromatographic analysis is the method’s sensitivity. Method validation is the systematic evaluation of an analytical procedure to demonstrate its scientific correctness under the conditions for which it is intended. In addition, the limit of detection (LOD) and the limit of quantification (LOQ) must be verified. These values are determined by the noise level and depend largely on the sensitivity of the analytical method [161,162]. The flame ionisation detector (FID) is considerably less sensitive than mass detectors. In the case of isobutanol and isopropanol, the LOD ranged from 0.2 to 1.01 mg/L and the LOQ from 0.65 to 3.38 mg/L, respectively [164]. The detection of the analytes by mass spectrometry (single-quadrupole) yielded lower LOD and LOQ values than FID detection, with values of 0.004 and 0.02 mg/L, respectively [151]. For triple-quadrupole mass spectrometry (QqQ-MS) analysis, LOD values of 0.05 μg/L were estimated [163]. In HS-SPME-GC×GC/TOFMS analysis, the LOD and LOQ values were low and dependent on the compound under investigation. The LODs ranged from 0.001 μg/L for ethyl isovalerate and hexanoic acid to 2554 μg/L for ethyl 3-hydroxybutanoate, while the LOQs ranged from 0.003 μg/L for ethyl isovalerate and hexanoic acid to 7582 μg/L for ethyl 3-hydroxybutanoate [161]. Lower values were also reported for wine thiol compounds analysed by chromatography–Orbitrap-mass spectrometry. The LOD was found to be between 0.002 and 0.012 μg/L, while the LOQ was between 0.006 and 0.043 μg/L [140].

The standards of tested compounds can be used to identify the compounds that shape the volatile profile of wines. However, due to the large number of compounds, it is practically impossible to determine all of them. Currently, the NIST library is used to identify these components through gas chromatography–mass spectrometry [169]. It is important to note that the mass spectra in the NIST library were primarily obtained using low-resolution mass spectrometry. Differences in fragment ions and ion abundances were observed between the high-resolution mass spectra obtained with GC-Orbitrap-MS and the low-resolution mass spectra obtained with GC-Quadrupole-MS, resulting in qualitative inaccuracy. To ensure accurate identification, it is necessary to establish high-resolution spectra (HRMS) of aromatic compounds analysed by GC-Orbitrap-MS [165].

When analysing volatile compounds that contribute to aroma, it is important to consider techniques for evaluating odour using the human sense of smell. One such technique is gas chromatography with olfactometry (GC-O), which employs the human nose as a detector to obtain data on odour-active compounds. This enables an understanding of the impact of these volatile substances on the specific aroma of wine [170,171,172]. The interactions between aroma compounds and other wine ingredients can influence the perception of aroma [173,174]. It is, however, important to note that the peak width in the GC×GC chromatographic system ranges from 30 to 140 ms, while the human respiration cycle lasts approximately 3–5 s. It is impossible to combine GC×GC with an olfactometric detector due to the discrepancies in the measurement timescales. Nevertheless, it can be performed using two-dimensional heart-cut gas chromatography (GC-GC). In GC-GC, a limited number of fractions of the eluate from the first column are transferred to a second column, after which, they are sniffed [168].

## 6. Conclusions

Fruit wines encompass a broad range of wine products, the aromatic profile of which is shaped by various factors. The type and species of fruit used to make the wine is the most important factor. Even within the same fruit species, a diverse range of wine aromas can be found, as the aroma is also influenced by the variety of fruit, growing and climatic conditions during the ripening period, and vinification techniques. The type and strain of yeast used for fermentation, and the fermentation and ageing conditions of the wines themselves, are also very important. It is possible to influence the volatiles profile of wines to a certain extent by modifying the temperature of fermentation and ageing. For example, lower temperatures facilitate the production of light fruity and floral esters, which are typically desirable in wines. Furthermore, the utilisation of lactic acid bacteria can exert a considerable influence on the aroma compound profile of fruit wines. The development of analytical techniques is facilitating the identification of numerous aroma components that are usually present in low concentrations. These components can influence the final bouquet of wines due to their low thresholds of sensory perception. The combination of an olfactometric analysis technique with gas chromatography coupled to mass spectrometry is of considerable importance. This enables the evaluation of both the qualitative and quantitative aspects of the compounds responsible for the aroma of wines. Additionally, it facilitates the identification of odours perceived by the human sense of smell, which are dependent on specific chemical compounds.

A novel approach to the analytical identification of volatile odour-active compounds in wines may be the electronic nose. Currently, devices employing mass spectrometry or metal oxide semiconductor sensors are available on the market. Although the complexity of the matrix of different wines does not yet provide satisfactory results, the development of artificial intelligence has the potential to accelerate the development of the electronic nose.

## Figures and Tables

**Table 1 molecules-29-02457-t001:** Higher alcohols of wine and their precursors [10,13,14].

Compound (IUPAC Name)	Amino Acid
Isobutanol (2-methylpropan-1-ol)	Valine
Isoamyl alcohol (3-methylbutan-1-ol)	Leucine
Tert-amyl alcohol (2-methylbutan-2-ol)	Isoleucine
Phenylethanol (2-phenylethanol)	Phenylalanine
Methionol (3-methylsulfanylpropan-1-ol)	Methionine

**Table 3 molecules-29-02457-t003:** Esters in wines, their aromas, and sensory detection thresholds.

Compound(IUPAC Name)	Chemical Structure	AromaDescriptor	Threshold [μg/L]	References
Ethyl acetate	** 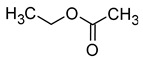 **	Apple/fruity	7500	[20]
Hexyl acetate	** 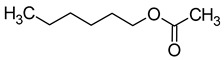 **	Apple/fruity	1500	[20]
Isoamyl acetate(3-methylbutyl acetate)	** 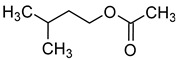 **	Banana	30	[39]
Isobutyl acetate(2-methylpropyl acetate)	** 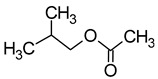 **	Solvent	1600	[24]
Phenylethyl acetate(2-phenylethyl acetate)	** 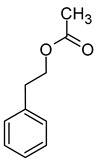 **	Roses	250	[39]
Ethyl cinnamate(ethyl (*E*)-3-phenylprop-2-enoate)	** 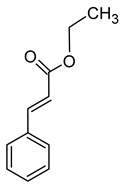 **	Cinnamate/sweet	1,1	[40]
Ethyl isobutyrate(ethyl 2-methylpropanoate)	** 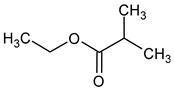 **	Fruity/strawberry	15	[39]
Ethyl lactate(ethyl 2-hydroxypropanoate)	** 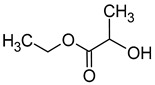 **	Fruity	154	[41]
Ethyl butyrate(ethyl butanoate)	** 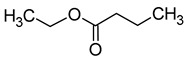 **	Fruity	20	[39]
Ethyl 2-methylbutyrate(ethyl 2-methylbutanoate)	** 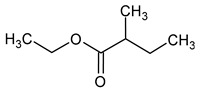 **	Fruity/green apple	1	[39]
Ethyl isovalerate(ethyl 3-methylbutanoate)	** 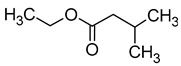 **	Fruity/anise	3	[40]
Ethyl 3-hydroxybutyrate(ethyl 3-hydroxybutanoate)	** 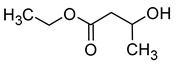 **	Apple/tropical fruity	21	[20,24]
Ethyl caproate(ethyl hexanoate)	** 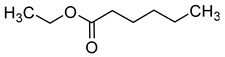 **	Fruity/anise	5	[39]
Ethyl caprylate(ethyl octanoate)	** 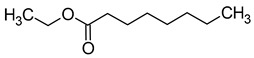 **	Fruity/fresh	2	[39]
Ethyl caprate(ethyl decanoate)	** 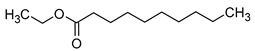 **	Fruity	200	[40]
Ethyl laurate(ethyl dodecanoate)	** 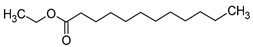 **	Floral/waxy	1500	[20]
Diethyl succinate(diethyl butanedioate)	** 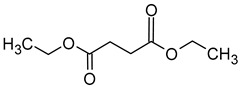 **	Fruity	200	[41]

Chemical structure based on Pubchem database [27].

**Table 4 molecules-29-02457-t004:** Fatty acids in wines, their aromas, and sensory detection thresholds.

Compound (IUPAC Name)	Chemical Structure	Aroma Descriptor	Threshold [µg/L]	References
Butyric acid (butanoic acid)	** 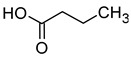 **	Cheese, butter	173	[20]
Caproic acid (hexanoic acid)	** 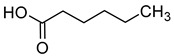 **	Cheese, fatty, baked potato	420	[20]
Isobutyric acid (2-methylpropanoic acid)	** 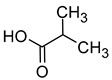 **	Cheese	2300	[40]
2-Methylbutyric acid (2-methylbutanoic acid)	** 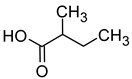 **	Cheese	33	[40]
Isovaleric acid (3-methylbutanoic acid)	** 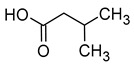 **	Cheese	33	[40]
Caprylic acid (octanoic acid)	** 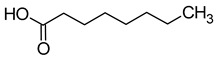 **	Fatty/unpleasant	500	[40]
Capric acid (decanoic acid)	** 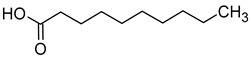 **	Cheese	1000	[40]

Chemical structure based on Pubchem database [27].

**Table 6 molecules-29-02457-t006:** Pyrazines in wines, their aromas, and sensory detection thresholds.

Compound (IUPAC Name)	Chemical Structure	Aroma Descriptor	Threshold [µg/L]	References
3-isopropyl-2-methoxypyrazine (2-methoxy-3-propan-2-ylpyrazine)	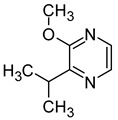	Asparagus, green beans	0.002	[64]
3-isobutyl-2-methoxypyrazine (2-methoxy-3-(2-methylpropyl)pyrazine)	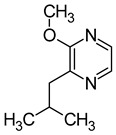	Green pepper	0.01–0.016	[64]
3-sec-butyl-2-methoxypyrazine (2-butan-2-yl-3-methoxypyrazine)	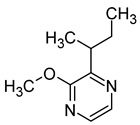	Pea	0.001–0.002	[64]

Chemical structure based on Pubchem database [27].

**Table 7 molecules-29-02457-t007:** Thiols in wines, their aromas, and sensory detection thresholds.

Compound (IUPAC Name)	Chemical Structure	Aroma Descriptor	Threshold [µg/L]	References
3-Mercapto-1-hexanol (3-sulfanylhexan-1-ol)	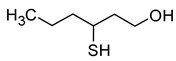	Grapefruit, citrus peel, passion fruit	0.06	[23]
3-Mercaptohexyl acetate (3-sulfanylhexyl acetate)	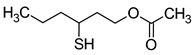	Passion fruit, box tree, boxwood	0.004	[23]
4-Mercapto-4-methyl-2-pentanone (4-methyl-4-sulfanylpentan-2-one)	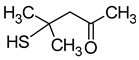	Blackcurrant, box tree, passion fruit	0.0008	[23]

Chemical structure based on Pubchem database [27].

**Table 8 molecules-29-02457-t008:** Compounds derived from wood that shape the aroma of wines (ranges based on the values reported in various scientific studies).

Compound (IUPAC Name)	Chemical Structure	Aroma Descriptor	Threshold [μg/L]	References
Furfural(2-Furancarboxaldehyde)	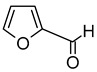	Sweet, woody, almond, bread	760–1500	[73,74]
5-Methylfurfural(5-Methyl-2-furancarboxaldehyde)	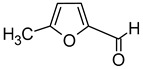	Caramel, breads, coffee, almond, burnt, sugar	1.1–1600	[73,75]
5-Hydroxymethylfurfural(5-Hydroxymethyl-2-furaldehyde)	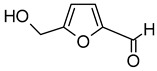	Vanilla, candy, caramel	10–10,000	[73,76]
*cis*-Whiskey lactone((4*R*,5*R*)-5-butyl-4-methyloxolan-2-one)	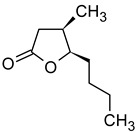	Woody, coconut, nutty, vanilla	6–46	[73,77]
*trans*-Whiskey lactone((4*S*,5*R*)-5-butyl-4-methyloxolan-2-one)	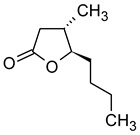	Spicy, coconut, clove, woody, vanilla	67–370	[73,74]
Guaiacol(2-Methoxyphenol)	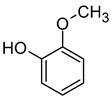	Clove, smoke, spice, sweet, medicine	9.5–20	[73,78]
Eugenol(2-Methoxy-4-(prop-2-enyl)Phenol)	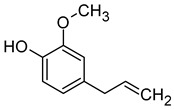	Clove, cinnamon, honey, spicy,	6	[73,74]
Isoeugenol(1-Methoxy-4-(prop-2-enyl) phenol)	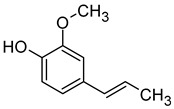	Spicy, floral, clove, woody	6	[24,73]
Vanillin(4-Hydroxy-3-Methoxybenzaldehyde)	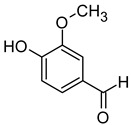	Vanilla, sweet	200–1000	[73,79]
Syringaldehyde(4-Hydroxy-3,5-dimethoxybenzaldehyde)	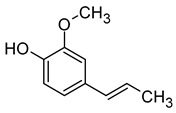	Woody, sweet, vanilla	50	[73,80]
o-Cresol(2-benzyl-4-chlorophenol)	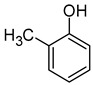	Leather/spicy	31	[81]

Chemical structure based on Pubchem database [27].

**Table 9 molecules-29-02457-t009:** The influence of apple varieties on the profile of volatile compounds in wines (W) and ciders (C) [mg/L] [83,87,88,89,91,102,103,104,105,106].

Compounds	Gloster (W)	Champion (W)	Idared (W)	Fuji (W)	Ilzer Rose (W)	Gala (C)	G. Delicious (C)	Mix (C)	D. de Tresali (C)
Acetaldehyde	1.4	6.9	6.4	na	na	na	21.47	na	na
Acetone	1.3	0.6	0.0	na	na	na	0.68	na	na
Ethyl acetate	5.8	27.0	30.1	54.8	0.32	na	6.79	17	na
Isoamyl acetale	0.7	na	na	16.66	1.02	0.14	0.11	0.39	1.74
Ethyl hexanoate	1.3	na	na	0.72	0.96	0.80	0.07	0.20	0.07
Methanol	18.8	121.0	20.0	na	na	na	na	35	na
Propanol	94.8	20.6	23.2	8.11	na	na	5.71	12	na
Isobutanol	237.8	95.0	110.0	184.85	0.44	0.05	7.34	32	na
Amyl alcohols	260.8	322.6	329.0	232.0	3.53	3.80	73.58	168	na
Butanol	na	20.2	27.1	8.46	na	na	na	na	na
Pentanol	na	1.3	1.4	0.06	0.04	na	na	0.04	na
Hexanol	na	5.2	6.2	2.18	0.43	na	na	0.04	0.59
Phenylethanol	na	38.4	44.2	43.5	0.63	0.02	na	17	na
Acetic acid	na	76.0	92.0	282.93	na	na	na	na	na
Ethyl butyrate	na	na	na	2.19	0.03	na	0.003	0.007	0.015
1-Butyl acetate	na	na	na	0.32	0.03	na	0.008	na	na
1-Hexyl acetate	na	na	na	0.32	1.07	0.05	na	na	na
Ethyl heptanoate	na	na	na	0.03	na	na	na	na	na
Ethyl lactate	na	na	na	4.63	na	na	na	na	na
Ethyl octanoate	na	na	na	1.09	2.08	0.53	0.12	0.18	0.95
Heptanol	na	na	na	0.07	0.01	na	na	na	na
2,3-Butanediol	na	na	na	0.06	0.05	0.17	0.08	na	na
1-Octanol	na	na	na	0.04	0.04	na	na	na	na
Methyl decanoate	na	na	na	0.01	0.04	na	na	na	na
Diethyl succinate	na	na	na	0.24	na	na	na	na	na
Ethyl decanoate	na	na	na	1.5	2.17	na	na	na	na
Methyl dec-4-enoate	na	na	na	0.01	0.01	na	na	na	na
3-Methylbutyl octanoate	na	na	na	0.01	0.10	na	na	na	na
Ethyl benzoate	na	na	na	0.99	0.00	na	na	na	na
Methyl laurate	na	na	na	1.27	na	na	na	na	na
Phenylethyl acetate	na	na	na	7.7	0.22	na	0.04	na	na
Ethyl laurate	na	na	na	112.29	na	na	na	na	na
Hexanoic acid	na	na	na	4.75	0.40	na	0.04	na	2.89
Ethyl myristate	na	na	na	2.71	na	na	na	na	na
Octanoic acid	na	na	na	6.11	0.72	na	0.06	na	2.62
Ethyl palmitate	na	na	na	15.1	na	na	na	na	na
Ethyl 9-hexadecenoate	na	na	na	16.37	na	na	na	na	na
Decanoic acid	na	na	na	2.41	na	na	0.01	na	1.93

na—not analysed.

**Table 11 molecules-29-02457-t011:** Compounds that contribute to the aroma of three different wines made from plum family fruits [mg/L].

Compound	W10 (Plum)	W11 (Apricot)	W12 (Peach)
Ethyl acetate	52.50	729.35	333.65
Isobutyl acetate	2.28	42.19	na
Ethyl butanoate	8.86	43.24	na
Butyl acetate	nd	4.88	na
Ethyl hexanoate	80.78	749.92	382.72
Hexyl acetate	1.34	60.09	93.01
Methyl octanoate	1.40	34.89	2.56
Ethyl octanoate	0.73	2552.69	1491.2
Isopentyl hexanoate	1.60	na	3.84
Ethyl nonanoate	5.46	44.32	8.11
Ethyl decanoate	1.11	1840.95	866.56
Isopentyl octanoate	10.64	na	29.44
Ethyl dodecanoate	192.00	175.20	na
Ethyl tetradecanoate	8.00	na	na
Ethyl hexadecanoate	7.50	na	na
Ethyl propionate	na	18.65	na
Ethyl isobutyrate	na	11.51	na
Propyl acetate	na	24.83	na
Isoamyl acetate	na	2472.35	223.16
Ethyl pentanoate	na	4.24	na
Ethyl heptanoate	na	9.26	na
Geranyl acetate	na	96.95	na
Ethyl benzoate	1.91	315.95	34.13
2-Phenylethyl acetate	na	42.38	na
1-Propanol	na	85.48	na
Isobutanol	na	159.27	301.65
Isoamyl alcohol	300.00	2605.68	1043.60
1-Hexanol	na	20.10	145.07
2,3-Butanediol	nd	14.70	na
1-Octanol	nd	13.62	na
1-Decanol	na	5.13	1.28
Benzyl alcohol	nd	20.07	10.67
Octanoic acid	10.50	na	57.17
Decanoic acid	15.00	na	na
Benzaldehyde	nd	na	5.97
2-Phenylethanol	28.56	201.16	67.41
Hexanoic acid	na	na	20.05
Eugenol	3.46	na	na
Linalool	na	731.90	23.47
α-Terpineol	na	135.46	1.28
Citronellol	na	18.48	3.41
Geraniol	na	40.89	na

Legend: W10—plum [123], W11—apricot [124], W12—peach [125]. na—not analysed, nd—not detected.

**Table 12 molecules-29-02457-t012:** Compounds that contribute to the aroma of four various types of berry wines [mg/L].

Compounds	W13 (Blueberry)	W14 (Raspberry)	W15 (Strawberry)	W16 (Mulberry)
Ethyl acetate	90.00	65.83	37.84	31.41
Ethyl butyrate	1.34	na	na	na
Isoamyl acetate	17.60	na	na	na
Ethyl isovalerate	0.198	na	na	na
Ethyl hexanoate	47.33	nd	5.32	11.71
Hexyl acetate	1.34	19.31	37.62	48.18
Ethyl caprylate	211.70	na	na	na
Ethyl caprate	335.20	na	na	na
Methyl salicylate	0.60	na	na	na
Ethyl phenylacetate	5.62	na	na	na
Ethyl laurate	165.00	na	na	na
Ethyl myristate	4.00	na	na	na
Ethyl octanoate	na	5.20	42.82	23.76
Ethyl decanoate	na	1.25	2.11	1.16
Diethyl succinate	na	4.06	1.20	1.46
2-Phenylethyl acetate	na	9.82	7.12	5.80
Methanol	2.40	na	na	na
1-Propanol	na	nd	3.20	nd
2-Methylpropanol	na	152.30	778.60	480.45
1-Butanol	na	nd	19.40	nd
3-Methylbutanol	na	3745.00	20,568.83	7885.75
1-Pentanol	na	20.00	nd	nd
1-Hexanol	8.00	5.03	25.21	51.73
1-Octanol	na	286.53	116.60	122.49
2-Phenylethyl alcohol	84.00	25.57	19.53	16.46
Acetic acid	na	163.63	67.08	69.68
2-Methylpropanoic acid	na	nd	8.79	12.21
Hexanoic acid	1.26	nd	nd	7.97
Octanoic acid	6.50	1.55	2.70	9.08
Decanoic acid	8.00	na	na	na
Lauric acid	1.00	na	na	na
Benzaldehyde	na	nd	1.22	19.39
Eugenol	3.72	na	na	na
Linalool	17.82	na	na	na
α-Terpineol	3.50	na	na	na

Legend: W13—blueberry [123], W14—raspberry [128], W15—strawberry [128], W16—mulberry [128]. na—not analysed. nd—not detected.

**Table 13 molecules-29-02457-t013:** Compounds that contribute to the aroma of citrus and pineapple wines [mg/L].

Compounds	W17 (Orange)	W18 (Orange)	W19 (Orange)	W20 (Ponkan Mandarin)	W21 (Pineapple)	W22 (Pineapple)
Ethyl acetate	25.28	na	na	4.36	na	0.31
Isoamyl acetate	101.83	0.50	0.60	18.51	1.67	1.97
Ethyl hexanoate	22.99	0.17	na	na	na	na
Hexyl acetate	na	na	0.01	na	na	na
Ethyl caprylate	0.58	na	na	na	na	na
Ethyl caprate	19.03	na	na	25.38	na	na
Ethyl 3-hexenoate	na	na	0.23	na	0.68	0.13
Ethyl lactate	na	1.77	0.09	na	0.93	na
Ethyl octanoate	na	0.30	0.11	31.26	1.03	0.33
Ethyl decanoate	5.40	0.29	0.02	na	2.89	0.06
Diethyl succinate	na	0.29	0.24	na	2.47	<0.01
2-Phenylethyl acetate	40.89	na	0.17	6.94	0.91	0.12
1-Propanol	na	1.16	na	na	na	na
2-Methylpropanol	na	6.20	1.37	na	2.56	0.21
1-Butanol	na	0.02	0.03	na	0.03	na
3-Methylbutanol	na	79.04	81.33	na	46.47	1.51
3-Methyl-3-buten-1-ol	na	0.03	0.01	na	na	na
1-Pentanol	0.91	0.04	0.03	53.95	0.80	na
1-Hexanol	na	0.25	0.51	na	0.06	na
2,3-Butanediol	na	1.80	na	na	na	0.02
1-Octanol	na	0.17	0.64	na	na	0.09
Benzyl alcohol	na	0.05	0.05	na	0.08	na
2-Phenylethyl alcohol	67.63	27.26	35.46	33.21	12.20	2.39
Acetic acid	na	na	na	na	na	0.21
Hexanoic acid	7.21	0.56	0.81	0.13	2.60	0.09
Capric acid	1.45	na	na	na	na	na
Octanoic acid	33.33	1.10	0.70	2.23	5.48	0.12
Decanoic acid	na	na	na	na	2.08	<0.1
Benzaldehyde	na	0.01	0.01	na	0.08	<0.01
Linalool	1.92	1.64	3.71	0.02	na	na
α-Terpineol	nd	0.84	1.62	na	na	na
β-Citronellol	1.01	0.14	0.15	na	na	na
Limonene	na	0.43	0.41	nd	na	0.01

Legend: W17—orange [136], W18—orange [138], W19—orange [137], W20—Ponkan mandarin [133], W21—pineapple [134], W22—pineapple [137]. na—not analysed, nd—not detected.

## Data Availability

Not applicable.

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
