# Peer review of "Volatilomics of Fruit Wines"

_molecules, 2024, doi:10.3390/molecules29112457_

Round 1

Reviewer 1 Report

Comments and Suggestions for Authors

The  review paper is ok

Author Response

You can find the answer to the review in the attachment.

Reviewer 2 Report

Comments and Suggestions for Authors

Title: Volatilomics of fruit wines

1. Journal: molecule

2. There are many grammatical errors in the text, and I hope that the author will seriously correct them or seek professional help.

3. Please replace the form with a three-line grid.

4. The method part needs to be refined.

5. The most important thing is that the repetition rate of this paper is too high, and it is necessary to doubt its scientific validity.

Comments on the Quality of English Language

None

Author Response

(The authors gave the same response as above.)

Reviewer 3 Report

Comments and Suggestions for Authors

This article looks into the area of volatilomics, specifically the volatile molecules that contribute to the odor of fruit wine.The review stresses the complex nature of the elements that influence the scent of fruit wines, as well as the importance of analytical approaches in analyzing volatile compounds. The article is well written, and it thoroughly summarizes the latest literature data. 
How to improve the article: 
- Stress the difference between different fruit wines and the difference between fruit wines and grape wines. Comparing the volatile compounds or the sensory qualities of various fruit wine categories may yield insightful information.
- Go deeper into methodological limitations, such as the preparation of the sample, detection limits, and interpretation of results. 
-Some parts of the article are in a textbook style, giving general information that can be found in textbooks. Since the work is really extensive, those parts should be shortened.
- The conclusion briefly mentions developments in analytical techniques but could elaborate on future research directions in the field of fruit wine volatilities.
-There are also some style and language comments (mostly in the introduction part), which are given in the attachment. 

Comments on the Quality of English Language

English should be checked. There are some spelling errors. 

Author Response

(The authors gave the same response as above.)

Round 2

Reviewer 2 Report

Comments and Suggestions for Authors

Accept in current form

Comments on the Quality of English Language

Accept in current form